# Adding Neoadjuvant Immunotherapy to Chemotherapy in Non-Metastatic Triple-Negative Breast Cancer: A Propensity-Matched Cohort Study from a Tertiary Cancer Center

**DOI:** 10.3390/cancers17243933

**Published:** 2025-12-09

**Authors:** Mahmoud Al-Masri, Yasmin Safi, Ramiz Kardan, Daliana Mustafa, Ola Ramadan, Rama AlMasri

**Affiliations:** 1Department of Surgery, King Hussein Cancer Center, Amman 11941, Jordan; ramiz.kardan@gmail.com (R.K.); or.16619@khcc.jo (O.R.); 2The Office of Scientific Affairs and Research, King Hussein Cancer Center, Amman 11941, Jordan; dalianamustafa@gmail.com; 3Department of Internal Medicine, King Hussein Cancer Center, Amman 11941, Jordan; rmalmasri97@gmail.com

**Keywords:** triple-negative breast cancer, pembrolizumab, neoadjuvant chemotherapy, pathological complete response, immunotherapy, surgical outcomes, immune-related adverse events, real-world evidence, survival

## Abstract

**Simple Summary:**

Triple-negative breast cancer is one of the most aggressive types of breast cancer, with limited treatment options and a high risk of recurrence. Recent studies have shown that adding pembrolizumab, a type of immunotherapy that helps the body’s immune system attack cancer, to standard chemotherapy can improve patient outcomes. However, most of this evidence comes from clinical trials, and little is known about how this treatment works in everyday clinical practice. In this study, we looked at real-world patients treated at our center to see whether the addition of pembrolizumab to chemotherapy improved outcomes and how safe and feasible it was. We found that patients who received pembrolizumab had higher rates of tumor disappearance before surgery, tolerated treatment well, and experienced manageable side effects. These results suggest that incorporating immunotherapy into standard care may help improve the survival and quality of life for patients with this challenging disease.

**Abstract:**

Background: Triple-negative breast cancer (TNBC) is an aggressive subtype with limited targeted treatment options. Immunotherapy has recently emerged as a potential strategy. The addition of pembrolizumab to neoadjuvant chemotherapy, as established in the KEYNOTE-522 trial, represents a major advancement in targeted immunotherapy for TNBC. However, real-world data validating its feasibility and outcomes remain limited. This study aims to evaluate, in real-life settings, the impact of adding pembrolizumab to neoadjuvant chemotherapy on complete pathological response (pCR), recurrence-free survival (RFS), and overall survival (OS) in patients with non-metastatic TNBC. Methods: This retrospective cohort study included patients treated at King Hussein Cancer Center (KHCC) between 2015 and 2022. Among 8523 breast cancer cases, 761 were TNBC. Eligible patients had non-metastatic TNBC, received neoadjuvant therapy, and underwent surgery. The immunotherapy group included patients treated with neoadjuvant pembrolizumab (2019–2022); the no-immunotherapy group received standard neoadjuvant chemotherapy (2015–2022). Propensity score matching (1:1, nearest neighbor) was performed based on pre-treatment covariates including age, BMI, clinical stage, comorbidities, smoking, and histopathology. Pathological response, complication rates, RFS, and OS were analyzed using logistic regression and Kaplan–Meier curves with log-rank testing. Results: The matched cohort included 130 patients (65 per group). The study groups’ baseline characteristics were well-balanced between the two groups. Postoperative complication rates were similar across groups, with no significant increase in adverse events observed in the immunotherapy group. The mean lymph node positivity ratio was significantly lower in the immunotherapy group (2.2 ± 7.7 vs. 24.3 ± 33.1, *p* < 0.001), indicating reduced nodal burden. Pathologic complete response (pCR) was markedly higher with immunotherapy (66.2% vs. 9.2%, *p* < 0.001). However, survival outcomes were significantly improved with immunotherapy. At three years, RFS was markedly higher in the immunotherapy group (91.8%; 95% CI: 85.0–99.0%) compared to the no-immunotherapy group (53.8%; 95% CI: 42.8–67.8%), with a log-rank *p* < 0.001. Overall survival also significantly favored the immunotherapy group, with three-year OS of 87.2% versus 67.8% in no-immunotherapy group (*p* = 0.0015). Conclusions: Neoadjuvant pembrolizumab significantly enhances pathological response, reduces nodal involvement, and provides durable RFS and OS benefits in non-metastatic TNBC without increasing perioperative complications. This study supports incorporating immunotherapy into standard neoadjuvant regimens for TNBC patients and provides real-world evidence from a Middle Eastern tertiary cancer center.

## 1. Introduction

Triple-negative breast cancer (TNBC) is a biologically aggressive subtype of breast malignancy, accounting for approximately 15–20% of all breast cancer diagnoses worldwide [1]. Defined by the absence of estrogen receptors (ERs), progesterone receptors (PRs), and human epidermal growth factor receptor 2 (HER2) amplification, TNBC is associated with an unfavorable clinical course, marked by high rates of early recurrence, visceral metastases, and poorer overall survival relative to other breast cancer subtypes [2,3,4]. Its intrinsic heterogeneity and lack of actionable molecular targets pose significant therapeutic challenges [5].

TNBC treatment traditionally relies on neoadjuvant chemotherapy (NACT) as the cornerstone approach for non-metastatic disease. NACT serves multiple purposes, including tumor downstaging in breast and axilla, facilitating breast-conserving surgery, and providing early systemic control of micrometastatic disease [6,7]. While standard NACT achieves relatively high rates of pathological complete response (pCR), which is associated with improved prognosis [8], a substantial subset of patients still experiences disease recurrence, particularly within the first few years post-treatment [9,10,11]. These suboptimal long-term outcomes underscore the limitations of chemotherapy alone in TNBC and highlight the urgent need for novel strategies to improve treatment efficacy and survival [5,12].

The immunogenic nature of TNBC has positioned it as a compelling candidate for immune-based therapies. This subtype is characterized by a higher prevalence of tumor-infiltrating lymphocytes (TILs), increased genomic instability, a higher tumor mutational burden (TMB), and elevated expression of programmed death-ligand 1 (PD-L1), all of which contribute to an active tumor immune microenvironment [13,14]. These features provide a strong rationale for the incorporation of immune checkpoint inhibitors (ICIs), such as pembrolizumab, into the neoadjuvant setting. Pivotal trials—most notably the KEYNOTE-522 study—have demonstrated that the addition of pembrolizumab to standard chemotherapy significantly enhances pCR rates and improves event-free survival (EFS) in early stage TNBC, establishing a new therapeutic paradigm [15,16,17]. At the 5-year follow-up, pembrolizumab plus chemotherapy was associated with a sustained event-free survival (EFS) benefit (81.2% vs. 72.2%) and an overall survival (OS) improvement (86.6% vs. 81.7%) compared to chemotherapy alone, thereby establishing a new therapeutic paradigm in early stage TNBC [18].

However, despite these encouraging findings, several important gaps remain in the literature. Most available data are derived from randomized controlled trials (RCTs) that employ stringent inclusion criteria and exclude patients with common real-world complexities, such as comorbid conditions or poor performance status. In addition, RCT participants typically demonstrate higher treatment adherence, standardized chemotherapy delivery, and closely monitored toxicity management compared with unselected clinical populations. Consequently, the generalizability of these results to everyday practice remains uncertain. Moreover, data on the real-world safety and surgical outcomes associated with neoadjuvant chemoimmunotherapy are limited [19,20]. Postoperative complications and immune-related adverse events in patients receiving combination regimens are particularly underreported, leaving an incomplete picture of the overall treatment burden [21,22]. Addressing this evidence gap, our study aimed to provide real-world validation of neoadjuvant pembrolizumab-based therapy in TNBC, assessing (1) pathological response rates, including pCR; (2) postoperative complication profiles to assess the safety and surgical outcomes of treatment; and (3) overall survival outcomes between patients receiving standard chemotherapy alone and those treated with chemo-immunotherapy.

## 2. Materials and Methods

### 2.1. Study Design and Patient Population

This retrospective cohort study included breast cancer patients treated at King Hussein Cancer Center (KHCC) between January 2015 and December 2022. A total of 8523 breast cancer cases were identified, of which 761 (8.93%) were triple-negative breast cancer (TNBC). From this subset, adult patients with non-metastatic TNBC who received neoadjuvant therapy and underwent definitive surgery were considered eligible for inclusion.

Patients who received neoadjuvant immunotherapy with pembrolizumab as part of their treatment protocol from 2019 onward were identified as the immunotherapy group. A comparator group of patients treated with standard neoadjuvant chemotherapy alone was selected from earlier years (2015–2019) and overlapping years (2019–2022).

Observational data from tertiary cancer centers, where patient populations are heterogeneous and treatment decisions reflect routine practice, can provide critical insights into the effectiveness, safety, and feasibility of integrating immunotherapy into standard care. In this context, the use of propensity score matching offers a robust methodological approach to reduce selection bias and adjust for baseline differences, thereby enhancing the comparability of treatment groups and mimicking the balance achieved in RCTs. To minimize confounding and ensure balance between groups, propensity score matching (PSM) was performed using a 1:1 nearest neighbor approach without replacement. Matching variables included age, body mass index (BMI), clinical stage at diagnosis, smoking status, comorbidity status, and tumor histopathology. After matching, the final analytical sample consisted of 130 patients: 65 in the immunotherapy group and 65 in the chemotherapy group (Figure 1).

### 2.2. Treatment Protocols

Neoadjuvant chemotherapy backbones followed institutional (KHCC) guidelines and were tailored according to tumor receptor status. For TNBC, the historical standard regimen consisted of anthracycline-based therapy (doxorubicin and cyclophosphamide [AC]) followed by a taxane (paclitaxel or docetaxel). With the adoption of the KEYNOTE-522 protocol, the backbone shifted to carboplatin plus paclitaxel, followed by anthracycline-based therapy, with pembrolizumab administered across all phases of treatment. Beginning in 2019, pembrolizumab was approved per institutional policy for patients younger than 65 years and was delivered concurrently with chemotherapy. Axillary management adhered to internal guideline recommendations. Pathologic complete response (pCR) was defined as absence of invasive carcinoma in breast and axillary specimens (ypT0/is, ypN0).

In the adjuvant setting, patients were offered continuation of pembrolizumab if they had received it during neoadjuvant therapy, or adjuvant capecitabine, as supported by the CREATE-X trial [23].

### 2.3. Outcomes Definition

The primary outcome of this study was pathological complete response (pCR), defined as the absence of residual invasive carcinoma in both the breast and axillary lymph nodes at the time of surgery (ypT0/Tis, ypN0). The lymph node ratio (LNR) was defined as the proportion of pathologically positive lymph nodes to the total number of excised nodes, expressed as a percentage.

Secondary outcomes included overall survival (OS), defined as the time from initiation of treatment to death from any cause, and recurrence-free survival (RFS), defined as the time from initiation of neoadjuvant therapy to the first documented disease progression, either local (LRFS) or distant recurrence (DRFS); patients who died without documented recurrence were censored at the date of death for RFS analysis. Safety outcomes were evaluated by documenting and grading immune-related adverse events (irAEs), as well as postoperative complications in both groups. Endocrine immune-related adverse events (irAEs) were monitored according to a standardized protocol. Patients underwent baseline thyroid function tests (TSH, free T4), cortisol, and blood glucose assessment prior to initiating immunotherapy. Follow-up labs were obtained before each dose of pembrolizumab.

Postoperative complications were assessed within 30 days of surgery and included overall postoperative complications, surgical site infection (SSI), hematoma, and wound dehiscence. Simple seromas were not included in the composite measure of overall postoperative complications but were instead reported independently as a distinct postoperative outcome.

### 2.4. Propensity Score Matching and Statistical Analysis

Propensity score matching (PSM) was conducted using 1:1 nearest neighbor matching without replacement. The model included the following covariates: age at diagnosis, body mass index (BMI), clinical stage (IA–IIIC), smoking status, comorbidity (yes/no), and histopathology (DCIS, IDC, ILC, IMC). Covariate balance was assessed using standardized mean differences (SMDs). As shown in the Love plot (see Appendix A [Figure A1]), the adjusted cohort demonstrated good balance across all covariates, with post-matching SMDs reduced to below 0.1 for all included variables. This indicates that the matching process effectively minimized the baseline differences between treatment groups. A total of 208 patients met the inclusion criteria, including 143 who received neoadjuvant chemotherapy alone and 65 who received neoadjuvant chemo-immunotherapy with pembrolizumab. Following 1:1 propensity score matching, 130 patients (65 in each group) were included in the final analytical cohort. Post-matching balance was assessed using standardized mean differences (SMDs), kernel density estimation, and jitter plots, with all covariates achieving an SMD < 0.1. All patients in the immunotherapy arm were matched successfully, while 78 unmatched patients from the chemotherapy group were excluded. Post-matching analysis confirmed adequate covariate balance across groups.

Primary endpoints were pathologic outcomes (pCR rates, nodal status, lymph node ratio) and safety (30-day postoperative complications). Tumor (T) and nodal (N) stages were analyzed as ordinal categorical variables. For visualization, they were numerically coded (e.g., T1 = 1, T2 = 2, etc.) to allow for violin plot representation. Kernel density estimation and jitter were applied to display data spread; hence, non-integer values on the *y*-axis do not indicate fractional staging. Secondary endpoints were recurrence-free survival (RFS) and overall survival (OS), estimated by Kaplan–Meier analysis and compared using the log-rank test. Patients who had not experienced the event of interest (death, or recurrence) by the date of last follow-up were considered censored at their last contact. For overall survival (OS), patients alive at the end of follow-up were censored at their last known alive date. For recurrence-free survival (RFS), patients without a recurrence (local or distant) were censored at their last follow-up visit. To account for the temporal difference between cohorts (2015–2019 vs. 2019–2022), follow-up began at the date of treatment initiation for all patients. Kaplan–Meier survival estimates incorporated censoring at last follow-up, ensuring valid comparisons despite differing follow-up durations. Median follow-up time was estimated using the reverse Kaplan–Meier method.

Odds ratios (ORs) for complications were calculated via univariable logistic regression. All tests were two-sided, and *p* < 0.05 was considered statistically significant. Analyses were performed using R software version 4.2.0.

## 3. Results

### 3.1. Baseline Characteristics Before and After Matching

Post-matching comparisons demonstrated no statistically significant differences between groups across all covariates, including age (*p* = 0.9051), BMI (*p* = 0.9131), smoking status (*p* = 0.6842), comorbidity (*p* = 0.8482), clinical stage (*p* = 0.8992), and histopathology (*p* = 0.6982). This indicates that the matching process effectively minimized the baseline imbalances between the two treatment groups.

For surgical management, there was no significant difference in the distribution of surgery types between the two groups (*p* = 0.6742), nor in ASA scores (*p* = 0.7932). However, significant differences were observed in pathological outcomes between the immunotherapy and no-immunotherapy groups. Sentinel lymph node biopsy (SLNBx) was more frequently performed in the immunotherapy group compared to the standard chemotherapy group (66.1% vs. 35.9%, *p* < 0.001), while axillary dissection (AD) was more common in the no-immunotherapy group (64.1% vs. 33.9%, *p* < 0.001) (Table 1).

### 3.2. Immunotherapy Induced Adverse Events

Immune-related adverse events (irAEs) were noted in 50.8% of patients, with the most common being hypothyroidism (35.4%). Other reported complications included adrenal insufficiency (9.2%), pneumonitis (3.1%), colitis (1.5%), and skin toxicity (1.5%) (Table 2).

### 3.3. Postoperative Complications Among the Study Groups

In the univariable logistic regression analysis comparing 30-day postoperative outcomes between patients who received neoadjuvant immunotherapy in addition to chemotherapy and those who received chemotherapy alone, a higher—though not statistically significant—odds of overall postoperative complications was observed in the immunotherapy group (OR = 1.41, 95% CI: 0.55–3.62, *p* = 0.48). There were no statistically significant differences between groups in terms of SSI (OR = 1.79, 95% CI: 0.61–5.25, *p* = 0.29), wound dehiscence (OR = 1.36, 95% CI: 0.29–7.12, *p* = 0.690), reoperation rate (OR = 1.00, 95% CI: 0.18–5.58, *p* = 1.000), or 30-day readmission (OR = 1.52, 95% CI: 0.24–11.87, *p* = 0.650). Similarly, the odds of seroma formation (OR = 0.72, 95% CI: 0.11–1.60, *p* = 0.43) and hematoma (OR = 1.00, 95% CI: 0.06–16.33, *p* = 0.99) did not differ significantly between the two groups. Collectively, these findings highlight that the overall postoperative complication profile was comparable (Figure 2).

### 3.4. Pathological Outcomes

Pathologic T staging also showed a marked difference, with a higher proportion of complete pathological response (ypT0) in the immunotherapy group (69.2%) versus 10.8% in the no-immunotherapy group (*p* < 0.001). Similarly, pathologic nodal staging (ypN0) was significantly more common in the immunotherapy group (88.7%) compared to the standard group (46.9%) (*p* < 0.001) (Figure 3).

Pathological complete response (pCR) rates differed significantly between the two groups (*p* < 0.001). In the no-immunotherapy group, 59 patients (90.8%) did not achieve pCR, while only 6 patients (9.2%) did. In contrast, in the immunotherapy group, 22 patients (33.8%) did not achieve pCR, whereas 43 patients (66.2%) achieved pCR. Moreover, the mean lymph node ratio (number of positive nodes to total examined) was significantly lower in the immunotherapy group (mean 2.2 ± 7.7) compared to the standard group (mean 24.3 ± 33.1; *p* < 0.001), reflecting a substantial reduction in nodal involvement (Table 3).

These findings suggest that the addition of immunotherapy enhances pathologic response at both the primary tumor and nodal levels.

### 3.5. Adjuvant Therapy Patterns

Among the 130 patients, the use of adjuvant radiotherapy did not significantly differ between groups (No-immunotherapy 75.4% vs. Immunotherapy 80.0%; *p* = 0.527). In contrast, the administration of adjuvant systemic therapy showed a marked difference, with only 24.6% of the No-immunotherapy group receiving systemic therapy compared to 83.1% in the Immunotherapy group (*p* < 0.001). When stratified by type of systemic therapy, immunotherapy was administered exclusively in the Immunotherapy group (78.5%), whereas chemotherapy was more common in the No-immunotherapy group (24.6% vs. 4.6%) (Table 4).


Survival outcomes—Overall survival outcomes

For the entire cohort, the median follow-up time was 36.15 months (interquartile range [IQR], 31.91–49.67 months), median follow-up was 74.2 months (IQR 22.9–76.1) for the No-Immunotherapy group, and 29.2 months (IQR 24.9–33.4) for the Immunotherapy group. The median overall survival was not reached during the follow-up period. The estimated 5-year overall survival rate was 65.0% (95% confidence interval [CI], 55.0–77.1%).

Following propensity score matching, Kaplan–Meier survival analysis demonstrated significantly improved overall survival in the immunotherapy group compared to the standard chemotherapy group (log-rank test, *p* = 0.0015). At 1 year, survival was identical in both groups at 98.5% (95% CI: 95.5–100.0%). However, divergence emerged at later time points: at 3 years, the immunotherapy group maintained a survival rate of 87.2% (95% CI: 74.8–100.0%), while the no-immunotherapy group declined to 67.8% (95% CI: 57.2–80.5%) (Figure 4).

### 3.6. Recurrence-Free Survival (RFS)

The Kaplan–Meier analysis showed a statistically significant improvement in recurrence-free survival among patients who received immunotherapy compared to those who received standard chemotherapy (log-rank test, *p* < 0.001). At 1 year, the RFS was 98.5% (95% CI: 95.5–100.0%) in the immunotherapy group versus 89.2% (95% CI: 82.0–97.1%) in the no-immunotherapy group. By 3 years, RFS remained high in the immunotherapy group at 91.8% (95% CI: 85.0–99.0%), while it dropped to 53.8% (95% CI: 42.8–67.8%) in the no-immunotherapy group. At 4 years, RFS was 81.6% (95% CI: 64.0–100.0%) for the immunotherapy group compared to 50.5% (95% CI: 39.4–64.6%) for the no-immunotherapy group. These findings suggest a clear benefit of adding immunotherapy in reducing disease recurrence over time (Figure 5).

In terms of distant recurrence-free survival (DRFS), this was significantly improved in the immunotherapy group compared to the no-immunotherapy group (*p* = 0.0018). At 3 years, DRFS in the immunotherapy group was 91.8% (95% CI: 85.0–99.0%) compared to 60.8% (95% CI: 49.4–74.9%) in the no-immunotherapy group. This difference persisted at 5 years, with DRFS rates of 81.6% versus 55.1%, respectively. These findings suggest that the addition of immunotherapy is associated with a reduced risk of distant metastasis (Figure 6a).

For locoregional recurrence (Figure 6b), the immunotherapy group showed significantly improved locoregional recurrence-free survival compared to the no-immunotherapy group. At 3 years, the immunotherapy group had an LRFS of 96.9% (95% CI: 92.8–100%) versus 85.6% (95% CI: 76.7–95.4%) in the no-immunotherapy group (*p* = 0.029). This trend remained consistent through 5 years of follow-up, suggesting that adding immunotherapy may enhance locoregional disease control in this patient population.

## 4. Discussion

In this propensity score-matched cohort study, we evaluated the impact of adding neoadjuvant immunotherapy to standard chemotherapy on pathological outcomes, postoperative complications, and survival in a TNBC. Matching effectively balanced baseline characteristics, minimizing confounding. Immune-related adverse events were observed in approximately half of the patients, predominantly hypothyroidism, highlighting the need for vigilant monitoring. Postoperative outcomes were comparable between groups, with no significant differences in surgical complications, reoperation, or 30-day readmission. Pathologically, the immunotherapy group demonstrated markedly improved responses, including higher rates of complete pathological response (ypT0), nodal clearance (ypN0), and lower lymph node ratios, reflecting enhanced tumor response. Adjuvant systemic therapy patterns differed substantially, with immunotherapy administered almost exclusively in the Immunotherapy arm. Survival analyses revealed a sustained long-term benefit, with significantly improved overall survival, recurrence-free survival, locoregional recurrence-free survival, and distant metastasis-free survival in the immunotherapy group, suggesting that the addition of checkpoint inhibition enhances both local and systemic disease control while maintaining an acceptable safety profile.

In our study, we found no significant differences in 30-day postoperative complications, including surgical site infections, wound dehiscence, reoperations, or readmissions, between patients who received neoadjuvant chemoimmunotherapy and those treated with chemotherapy alone. These findings are consistent with those reported by Myers et al., who compared 139 patients treated with the KEYNOTE-522 regimen to 287 patients who received chemotherapy alone. They observed postoperative complications in 7.9% of the chemoimmunotherapy group versus 9.1% of the chemotherapy group (*p* = 0.6), with no significant differences in the need for reoperation (1.4% vs. 2.1%, respectively). While the most common complication in their chemotherapy cohort was surgical site infection and abscess, ischemia of mastectomy flaps was more frequent in the chemoimmunotherapy cohort, though still uncommon [24]. Taken together, these results reinforce that the addition of immunotherapy to standard chemotherapy does not increase short-term surgical morbidity, supporting the safety of integrating immune checkpoint inhibitors into neoadjuvant treatment for triple-negative breast cancer.

In addition to evaluating surgical outcomes, Myers et al. also reported on the frequency and type of immune-related adverse events (irAEs) among patients treated with the KEYNOTE-522 regimen. In the KN-522 cohort, 43% of patients experienced immune-related adverse events (irAEs), with hypothyroidism (15.9%), hepatitis (14.6%), adrenal insufficiency (9.8%), dermatitis (9.8%), and pneumonitis (8.5%) being the most common toxicities. In contrast, our study demonstrated a higher overall incidence of irAEs (50.8%), with endocrine toxicities—particularly hypothyroidism—being disproportionately more frequent (35.4% vs. 15.9% in KN-522). Other complications such as adrenal insufficiency (9.2% vs. 9.8%) and colitis (1.5% vs. 4.9%) were comparable or slightly lower, while pneumonitis (3.1% vs. 8.5%) and dermatologic toxicity (1.5% vs. 9.8%) were less frequent in our cohort [24]. These differences may reflect variations in baseline patient characteristics, disease stage, genetic or ethnic predispositions, as well as differences in detection and reporting practices. Importantly, the higher burden of endocrine toxicities in our cohort underscores the need for vigilant long-term monitoring and tailored supportive care in patients receiving chemoimmunotherapy.

In terms of treatment response, our findings parallel and extend the results reported by Helal et al., who recently investigated predictors of pathological complete response (pCR) in patients with stage II–III triple-negative breast cancer (TNBC) treated with neoadjuvant chemoimmunotherapy (NACi) at Institut Curie hospitals. In their cohort of 208 patients, the overall pCR rate was 70%, consistent with other reports of NACi efficacy in TNBC. Similarly, in our study, the addition of immunotherapy to neoadjuvant chemotherapy yielded a pCR rate of 66.2%, closely mirroring the outcomes reported by Helal et al. and confirming the reproducibility of this benefit across independent populations [25].

Notably, however, our chemotherapy-only group achieved a markedly lower pCR rate of 9%. The relatively low pathological complete response (pCR) rate observed in our chemotherapy-only group was lower than the rates commonly reported in the literature, which typically range from 15% to 50% in TNBC cohorts [26,27,28]. Several factors may explain this discrepancy. First, our cohort included a clinically advanced population with higher baseline tumor burden and nodal involvement, which are known to negatively impact response to chemotherapy [29]. Second, differences in chemotherapy regimens, dose intensity, and treatment adherence could have contributed to suboptimal responses [30]. Third, biological heterogeneity, including variations in tumor-infiltrating lymphocytes (TILs), PD-L1 expression, and tumor mutational burden, may have influenced chemosensitivity [28,31]. Lastly, the relatively small sample size of our chemotherapy group (n = 65) increases the impact of individual patient outcomes on overall pCR rates, potentially amplifying apparent differences compared to larger clinical trials. These considerations highlight the importance of patient selection and underscore the potential benefit of adding immunotherapy to enhance response rates in TNBC populations.

Our findings are consistent with the KEYNOTE-522 trial, which showed that adding pembrolizumab to neoadjuvant chemotherapy significantly improved the pathological complete response (pCR) rates and event-free survival in triple-negative breast cancer (TNBC). In KEYNOTE-522, 1174 patients were randomized, with a median follow-up of 39.1 months, and the 36-month event-free survival was 84.5% in the pembrolizumab–chemotherapy group versus 76.8% in the placebo–chemotherapy group (hazard ratio, 0.63; *p* < 0.001), with adverse events primarily occurring during the neoadjuvant phase and consistent with known safety profiles [32]. In our cohort, the addition of immunotherapy similarly resulted in markedly improved pathological responses, with higher rates of complete tumor response (ypT0, 69.2% vs. 10.8%) and nodal clearance (ypN0, 88.7% vs. 46.9%) compared to chemotherapy alone. Immune-related adverse events in our population were common (50.8%) but largely endocrine in nature and manageable, which was higher than that reported in KEYNOTE-522 (35%), further supporting the tolerability of neoadjuvant PD-1 blockade [32]. Overall, these results reinforce both the efficacy and safety of incorporating immunotherapy into neoadjuvant regimens for TNBC in clinical practice.

Our findings are also consistent with the results of the phase II NeoPACT trial, which evaluated an anthracycline-free neoadjuvant regimen of pembrolizumab plus carboplatin and docetaxel. In that study, a pCR rate of 58% and a 3-year event-free survival (EFS) of 86% were reported, demonstrating that effective tumor clearance and durable survival benefits can be achieved even in the absence of anthracyclines. While the NeoPACT trial highlights the feasibility of a less cardiotoxic, anthracycline-sparing approach, our cohort—treated predominantly with anthracycline-based chemoimmunotherapy—achieved even higher rates of ypT0 (69.2%) and nodal clearance (88.7%), as well as a 5-year overall survival of 87.2% [33].

Together, these results reinforce that pembrolizumab is a key driver of therapeutic efficacy in TNBC, with flexibility in chemotherapy backbone selection, while also underscoring the potential for regimen tailoring to optimize both efficacy and long-term safety. However, these findings should be interpreted with caution. Our cohort differs from KEYNOTE-522 and NeoPACT in several ways, including patient selection, treatment era, and outcome reporting. The chemotherapy-only group had a lower pCR rate, likely influenced by a higher proportion of advanced-stage disease, incomplete chemotherapy delivery data, and the absence of biomarker stratification (e.g., PD-L1, TMB, or AR status). Historical differences in adjuvant therapy use may have also contributed to the observed survival differences. Combined with the modest sample size, these factors may partly explain the unusually large effect sizes observed in both pCR and survival outcomes.

While our results support the benefit of neoadjuvant chemoimmunotherapy, confirmation in larger, multi-institutional cohorts with comprehensive treatment and biomarker data is warranted.

### Strengths and Limitations

The major strengths of this study include its real-world matched cohort design, which allowed for a robust comparison between patients receiving standard chemotherapy and those treated with chemoimmunotherapy. Propensity score matching ensured balanced groups and minimized baseline differences, while the relatively long follow-up period enabled the evaluation of multiple survival endpoints.

Nonetheless, several limitations must be acknowledged. This was a single-institution dataset with a modest sample size of 130 patients, which may reduce the generalizability of the findings and limit statistical power in subgroup analyses. Furthermore, the absence of biomarker stratification—such as PD-L1 expression, tumor mutational burden (TMB), or tumor-infiltrating lymphocytes (TILs)—restricts the ability to identify subgroups that may derive differential benefit from immunotherapy.

Detailed data on chemotherapy delivery, including cumulative dose intensity, number of completed cycles, and treatment delays, were not consistently available in archived records, limiting the assessment of their impact on pathologic response. Additionally, androgen receptor (AR) status was not included as a matching variable or adjusted covariate, although AR-positive TNBC is known to demonstrate lower pathologic complete response (pCR) rates compared with AR-negative disease. Both chemotherapy dose intensity and AR status will be incorporated into our planned five-year follow-up analysis to enable a more comprehensive evaluation of treatment response and survival predictors.

Detailed characterization of immune-related adverse events, including CTCAE severity grades, timing of onset, and management strategies, was not available and will be addressed in future studies. Furthermore, multivariable analyses for surgical outcomes and postoperative complications, adjusting for procedure type, extent of axillary surgery, and comorbidities, were not performed and will be addressed in future studies.

The lower rate of adjuvant systemic therapy observed in the non-immunotherapy group (24.6%) likely reflects temporal differences in treatment standards, as patients treated before the integration of the CREATE-X trial findings were less likely to receive adjuvant chemotherapy. This historical variation may have contributed to the observed imbalance between groups; however, adjuvant therapy receipt will be included as a matching variable in the forthcoming five-year survival analysis to minimize this potential confounding effect.

Residual Cancer Burden (RCB) classification was not routinely available for all patients, which may limit the precision of post-neoadjuvant pathologic response assessment. Future work incorporating RCB scoring will enable a more detailed correlation between treatment response and survival outcomes. While Kaplan–Meier and log-rank analyses were used in this study, multivariable analyses adjusting for residual confounders were not performed and should be considered in future work to strengthen causal inference. Finally, the observed difference in lymph node ratio (LNR) between groups likely reflects procedural variability in axillary management, as patients in the immunotherapy group underwent fewer axillary lymph node dissections and more sentinel lymph node biopsies.

## 5. Conclusions

Adding immunotherapy to neoadjuvant chemotherapy in this cohort significantly enhanced pathological responses, with higher rates of complete tumor and nodal clearance and a reduced need for axillary interventions. These improvements translated into substantial long-term survival benefits, including superior overall, recurrence-free, locoregional recurrence-free, and distant metastasis-free survival compared to chemotherapy alone. Importantly, the addition of immunotherapy did not increase the postoperative complication rates, supporting its safety in the perioperative setting. Immune-related adverse events were observed in a notable proportion of patients but were generally manageable, predominantly affecting endocrine function.

Future research should focus on validating these findings in larger, prospective datasets to determine whether the observed trend toward surgical de-escalation—specifically the increased feasibility of sentinel lymph node biopsy over axillary dissection—can be consistently reproduced. Incorporating biomarker analysis, including PD-L1 status, tumor mutational burden (TMB), and BRCA mutations, will be essential to refine patient selection and optimize therapeutic benefit. Moreover, survivorship strategies must extend beyond oncologic outcomes to address long-term immune-related adverse effects, particularly endocrine toxicities such as thyroid and adrenal dysfunction, which may impact quality of life long after treatment completion.

## Figures and Tables

**Figure 1 cancers-17-03933-f001:**
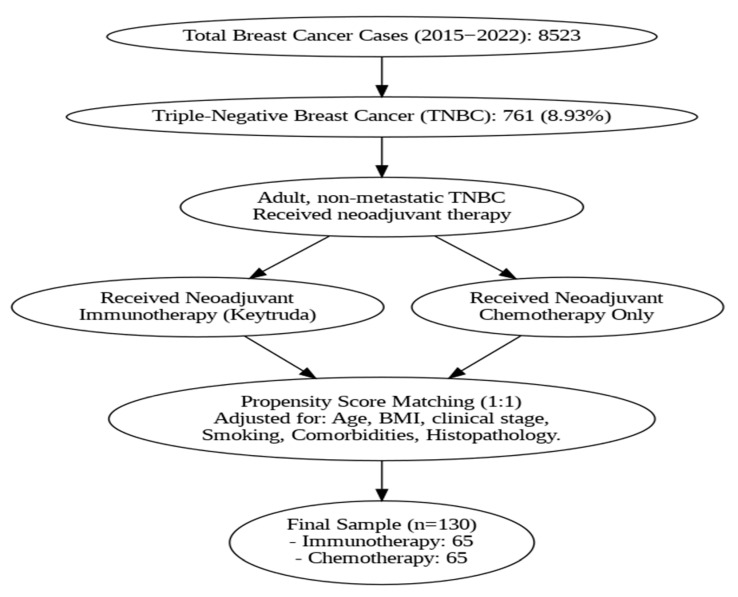
Study sample flowchart.

**Figure 2 cancers-17-03933-f002:**
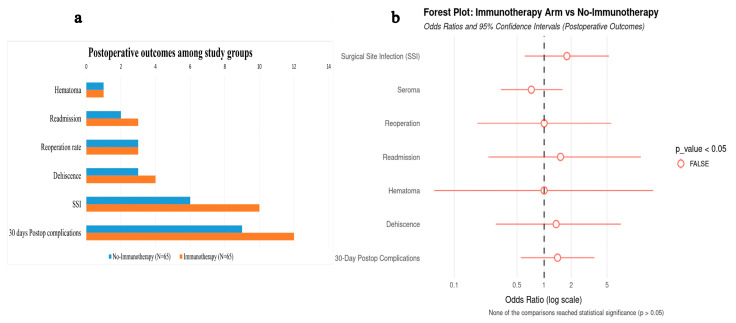
Postoperative outcomes among immunotherapy and no-immunotherapy groups. (**a**) Bar chart showing the frequency of specific postoperative outcomes in the immunotherapy (orange) and no-immunotherapy (blue) groups (N = 65 per group). (**b**) Forest plot depicting the odds ratios (ORs) and 95% confidence intervals (CIs) for postoperative outcomes comparing the immunotherapy arm to the no-immunotherapy arm.

**Figure 3 cancers-17-03933-f003:**
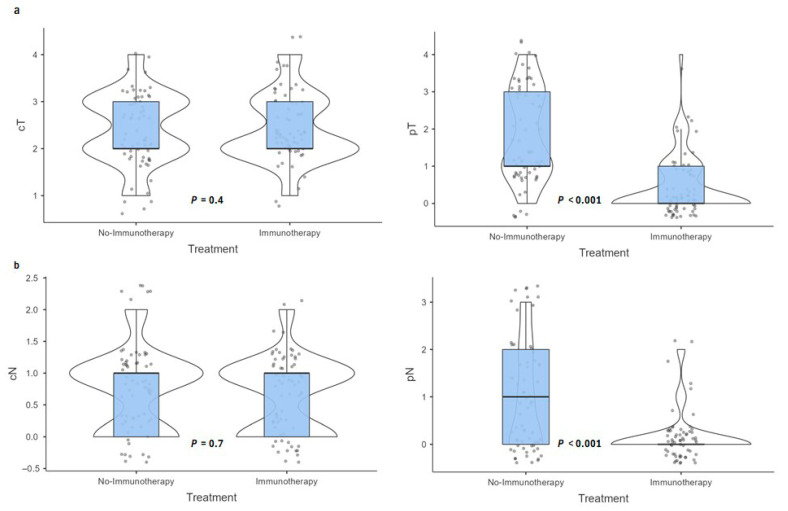
Comparison of Clinical and Pathological (**a**) T and (**b**) N Staging Between Immunotherapy and No-Immunotherapy Groups. The violin plots illustrate the data distribution, while the embedded box plots display the median and interquartile range.

**Figure 4 cancers-17-03933-f004:**
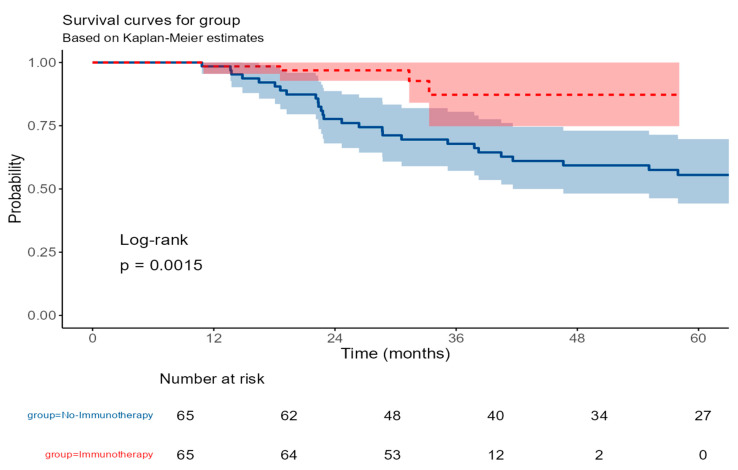
Kaplan–Meier curves for overall survival in patients receiving immunotherapy versus no immunotherapy. The red line represents the immunotherapy group, and the blue line represents the no-immunotherapy group. Shaded areas around the curves indicate the 95% confidence intervals.

**Figure 5 cancers-17-03933-f005:**
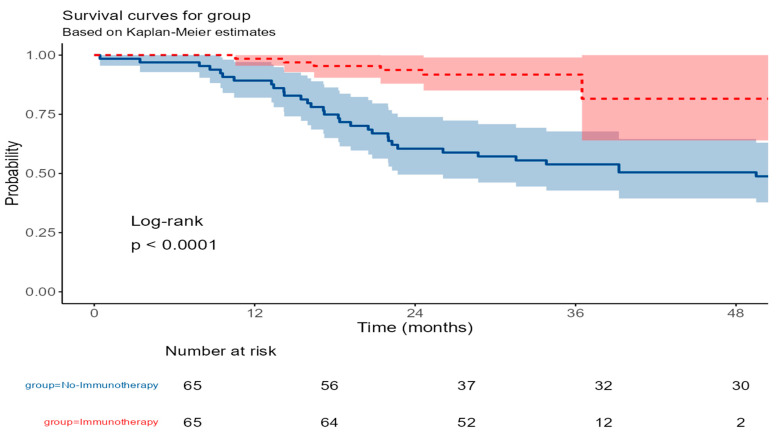
Kaplan–Meier curves for recurrence-free survival in patients receiving immunotherapy versus no immunotherapy. The red line represents the immunotherapy group, and the blue line represents the no-immunotherapy group. Shaded areas around the curves indicate the 95% confidence intervals.

**Figure 6 cancers-17-03933-f006:**
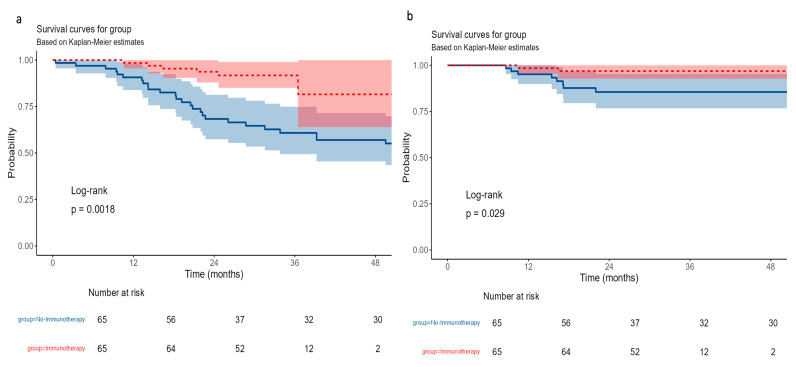
Kaplan–Meier curves for recurrence-free survival following propensity score matching. (**a**) Distant recurrence. (**b**) Locoregional recurrence. The red line represents the immunotherapy group, and the blue line represents the no-immunotherapy group. Shaded areas around the curves indicate the 95% confidence intervals.

**Table 1 cancers-17-03933-t001:** Baseline demographic and clinical characteristics of patients receiving neoadjuvant chemotherapy versus chemoimmunotherapy before and after propensity score matching (PSM).

	Pre-PSM			Post-PSM		
	No-Immunotherapy (N = 143)	Immunotherapy (N = 65)	*p*	No-Immunotherapy (N = 65)	Immunotherapy (N = 65)	*p*
Age at diagnosis			0.027 ^1^			0.905 ^1^
Mean (SD)	48.9 (12.7)	44.9 (10.0)		45.2 (11.8)	44.9 (10.0)	
Range	19.0–84.0	22.0–68.0		19.0–65.0	22.0–65.0	
BMI			0.608 ^1^			0.913 ^1^
Mean (SD)	31.0 (6.5)	30.5 (6.6)		30.4 (6.0)	30.5 (6.6)	
Range	20.0–50.3	19.6–56.7		20.0–45.0	19.6–56.7	
Smoking Status			0.592 ^2^			0.684 ^2^
No	105.0 (73.4%)	50.0 (76.9%)		48.0 (73.8%)	50.0 (76.9%)	
Yes	38.0 (26.6%)	15.0 (23.1%)		17.0 (26.2%)	15.0 (23.1%)	
Comorbidity			0.244 ^2^			0.848 ^2^
No	87.0 (60.8%)	45.0 (69.2%)		46.0 (70.8%)	45.0 (69.2%)	
Yes	56.0 (39.2%)	20.0 (30.8%)		19.0 (29.2%)	20.0 (30.8%)	
Clinical T			0.086 ^2^			0.349 ^2^
1	19.0 (13.3%)	4.0 (4.6.1%)		7.0 (10.8%)	4.0 (6.2%)	
2	76.0 (53.1%)	35.0 (53.8%)		27.0 (41.5%)	35.0 (53.8%)	
3	43.0 (30.1%)	20.0 (30.8%)		27.0 (41.5%)	20.0 (30.8%)	
4	5.0 (3.5%)	6.0 (9.2%)		4.0 (6.2%)	6.0 (9.2%)	
Clinical N			0.516 ^2^			0.742 ^2^
0	65.0 (45.5%)	26.0 (40.0%)		23.0 (35.4%)	26.0 (40.0%)	
1	64.0 (44.8%)	35.0 (53.8%)		36.0 (55.4%)	35.0 (53.8%)	
2	12.0 (8.4%)	4.0 (6.2%)		6.0 (9.2%)	4.0 (6.2%)	
3	2.0 (1.4%)	0.0 (0.0%)		-	-	
Clinical stage			0.302 ^2^			0.899 ^2^
1A	14.0 (9.8%)	3.0 (4.6%)		4.0 (6.2%)	3.0 (4.6%)	
2A	42.0 (29.4%)	16.0 (24.6%)		13.0 (20.0%)	16.0 (24.6%)	
2B	48.0 (33.6%)	22.0 (33.8%)		25.0 (38.5%)	22.0 (33.8%)	
3A	32.0 (22.4%)	18.0 (27.7%)		19.0 (29.2%)	18.0 (27.7%)	
3B	5.0 (3.5%)	6.0 (9.2%)		4.0 (6.2%)	6.0 (9.2%)	
3C	2.0 (1.4%)	0.0 (0.0%)		-	-	
Grade			0.498 ^2^			0.329 ^2^
1	3.0 (2.1%)	0.0 (0.0%)		1.0 (1.5%)	0.0 (0.0%)	
2	25.0 (17.5%)	12.0 (18.5%)		17.0 (26.2%)	12.0 (18.5%)	
3	115.0 (80.4%)	53.0 (81.5%)		47.0 (72.3%)	53.0 (81.5%)	
Histopathology			0.602 ^2^			0.698 ^2^
DCIS	2.0 (1.4%)	0.0 (0.0%)				
IDC	130.0 (90.9%)	61.0 (93.8%)		62.0 (95.4%)	61.0 (93.8%)	
ILC	2.0 (1.4%)	0.0 (0.0%)				
IMC	9.0 (6.3%)	4.0 (6.2%)		3.0 (4.6%)	4.0 (6.2%)	
Surgery			0.116 ^2^			0.674 ^2^
BCS	65.0 (45.5%)	23.0 (35.4%)		20.0 (30.8%)	23.0 (35.4%)	
Mastectomy with reconstruction	21.0 (14.7%)	17.0 (26.2%)		15.0 (23.1%)	17.0 (26.2%)	
Mastectomy without reconstruction	57.0 (39.9%)	25.0 (38.5%)		30.0 (46.2%)	25.0 (38.5%)	
Axillary surgery			<0.001 ^2^			<0.001 ^2^
AD	86 (60.1%)	21 (33.9%)		41 (64.1%)	21 (33.9%)	
SLNBx	57 (40.6%)	41 (66.1%)		23 (35.9%)	41 (66.1%)	
ASA			0.475 ^2^			0.793 ^2^
1	1 (0.7%)	0.0 (0.0%)				
2	134.0 (93.7%)	60.0 (92.3%)		61.0 (93.8%)	60.0 (92.3%)	
3	7.0 (4.9%)	5.0 (7.7%)		4.0 (6.1%)	5.0 (7.7%)	
4	1 (0.7%)	0.0 (0.0%)				

*p*-values were calculated using ^1^ Student’s *t*-test for continuous variables and ^2^ chi-square test or Fisher’s exact test for categorical variables. PSM = Propensity Score Matching; SD = Standard Deviation; BMI = Body Mass Index; DCIS = Ductal Carcinoma in Situ; IDC = Invasive Ductal Carcinoma; ILC = Invasive Lobular Carcinoma; IMC = Invasive Mammary Carcinoma; BCS = Breast-Conserving Surgery; AD = Axillary Dissection; SLNBx = Sentinel Lymph Node Biopsy; ASA = American Society of Anesthesiologists.

**Table 2 cancers-17-03933-t002:** Immune-related adverse events (N = 65).

Immune-Related Adverse Event	N (%)
No	32 (49.2%)
Yes	33 (50.8%)
If yes, what is the type	
Hypothyroidism	23 (35.4%)
Adrenal insufficiency	6 (9.2%)
Colitis	1 (1.5%)
Skin toxicity	1 (1.5%)
Pneumonitis	2 (3.1%)

**Table 3 cancers-17-03933-t003:** Pathological Outcomes by Treatment Group (Immunotherapy vs. No-Immunotherapy).

Outcome	No-Immunotherapy (N = 65)	Immunotherapy (N = 65)	*p*-Value
pCR			<0.001 ^1^
No	59 (90.8%)	22 (33.8%)	
Yes	6 (9.2%)	43 (66.2%)	
Pathologic T Stage (pT)			<0.001 ^1^
pT4	7 (10.8%)	1 (1.5%)	
pT3	18 (27.7%)	0 (0.0%)	
pT2	7 (10.8%)	5 (7.7%)	
pT1	26 (40.0%)	14 (21.5%)	
pT0	7 (10.8%)	45 (69.2%)	
Pathologic N Stage (pN)			<0.001 ^1^
pN0	30 (46.9%)	55 (88.7%)	
pN1	13 (20.3%)	4 (6.5%)	
pN2	12 (18.8%)	3 (4.8%)	
pN3	9 (14.1%)	0 (0.0%)	
LN positivity ratio			<0.001 ^2^
Mean (SD)	24.3 (33.1)	2.2 (7.7)	
Range	0.0–100.0	0.0–41.2	

*p*-values were calculated using ^1^ the chi-square test or Fisher’s exact test for categorical variables and ^2^ Student’s *t*-test for continuous variables, as appropriate. SD = Standard Deviation; LN = Lymph Node; pCR = Pathologic Complete Response.

**Table 4 cancers-17-03933-t004:** Distribution of Radiotherapy and Systemic Therapy Between Study Groups.

	No-Immunotherapy (N = 65)	Immunotherapy (N = 65)	Total (N = 130)	*p* Value
Adjuvant Radiotherapy				0.527 ^1^
No	16.0 (24.6%)	13.0 (20.0%)	29.0 (22.3%)	
Yes	49.0 (75.4%)	52.0 (80.0%)	101.0 (77.7%)	
Adjuvant systemic therapy				<0.001 ^1^
No	49.0 (75.4%)	11.0 (16.9%)	60.0 (46.2%)	
Yes	16.0 (24.6%)	54.0 (83.1%)	70.0 (53.8%)	
Type of systemic therapy				-
Immunotherapy	0.0 (0.0%)	51.0 (78.5%)	51.0 (39.2%)	
Chemotherapy	16.0 (24.6%)	3.0 (4.6%)	19.0 (14.6%)	

^1^ Pearson’s Chi-squared test.

## Data Availability

The data that support the findings of this study are available from the corresponding author upon reasonable request. Due to the retrospective nature of the study and the use of anonymized patient records, data are not publicly available to protect patient privacy and comply with ethical restrictions.

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
