# Peer review of "Adding Neoadjuvant Immunotherapy to Chemotherapy in Non-Metastatic Triple-Negative Breast Cancer: A Propensity-Matched Cohort Study from a Tertiary Cancer Center"

_cancers, 2025, doi:10.3390/cancers17243933_

Round 1

Reviewer 1 Report

Comments and Suggestions for Authors

It is a good job without mistyping 

Only in table 1 number of patients was 143, and I haven't seen it anywhere in the text

and getting confused in control groups, and I prefer to be as mentioned many times as neoadjuvant chemotherapy a corresponding reference group 

Author Response

Reviewer 1

  • Only in table 1 number of patients was 143, and I haven't seen it anywhere in the text

Response: We thank the reviewer for this valuable observation. In total, 208 patients with non-metastatic TNBC who received neoadjuvant therapy were initially identified, of whom 143 received neoadjuvant chemotherapy alone and 65 received neoadjuvant chemo-immunotherapy with pembrolizumab. To minimize confounding, 1:1 propensity score matching was applied, resulting in a final analytical sample of 130 patients (65 per group). We have now clarified this information in the Methods section to ensure consistency with Table 1.

Methods - Propensity Score Matching and Statistical Analysis:

“A total of 208 patients met the inclusion criteria, including 143 who received neoadju-vant chemotherapy alone and 65 who received neoadjuvant chemo-immunotherapy with pembrolizumab. Following 1:1 propensity score matching, 130 patients (65 in each group) were included in the final analytical cohort.”

  • and getting confused in control groups, and I prefer to be as mentioned many times as neoadjuvant chemotherapy a corresponding reference group 

Response: We appreciate the reviewer’s comment and agree that consistent terminology enhances clarity. Throughout the revised manuscript, we now consistently refer to the comparator group as the “no-immunotherapy group” instead of using varying terms such as “control group,”.

Reviewer 2 Report

Comments and Suggestions for Authors

The study by Al-Masri et al evaluated the effect of adding pembrolizumab to neoadjuvant chemotherapy in triple negative breast cancer. The authors used matched cohorts (1:1, n=65 each) of real-world patients treated with neoadjuvant chemotherapy vs neoadjuvant chemotherapy+immunotherapy from a single institution to compare pathological complete response (primary outcome), overall and disease-free survival. 

The study have found that the addition of immunotherapy to chemotherapy was well tolerated since immune-related adverse effects were manageable and no significant increase in post-surgical complications was detected compared to the group treated solely with chemotherapy. Further, pathological complete response was significantly higher in immunotherapy-treated group, as well as overall survival and relapse-free survival. Overall the study brings an important contribution to the field. 

The only issue to be clarified relates to figure 4: please better describe what is depicted and discuss this result, since staging could not assume non-integer values as depicted in the graphs.

Author Response

Reviewer 2

  1. The only issue to be clarified relates to figure 4: please better describe what is depicted and discuss this result, since staging could not assume non-integer values as depicted in the graphs.

Response: We thank the reviewer for this valuable comment and fully agree that staging variables (cT, pT, cN, and pN) are categorical and cannot take non-integer values. The non-integer appearance in Figure 4 results from the visualization method used (violin plots with kernel density estimation and jitter), which numerically coded the categorical stages (e.g., T1 = 1, T2 = 2, etc.) to display distribution and density.

To clarify this, we have revised the Figure 4 legend and Methods section to explicitly state that non-integer values are a graphical artifact and do not represent fractional staging. We have also expanded the Results description to highlight the significant pathological downstaging (pT and pN) observed in the immunotherapy group (P < 0.001).

Changes made:

  • Added explanation in the Statistical Analysis subsection.

Methods - Propensity Score Matching and Statistical Analysis:

Tumor (T) and nodal (N) stages were analyzed as ordinal categorical variables. For visualization, they were numerically coded (e.g., T1 = 1, T2 = 2, etc.) to allow violin plot representation. Kernel density estimation and jitter were applied to display data spread; hence, non-integer values on the y-axis do not indicate fractional staging.”

Reviewer 3 Report

Comments and Suggestions for Authors

Overall assessment

This is a timely and clinically relevant real-world study assessing the addition of neoadjuvant pembrolizumab to chemotherapy in early-stage TNBC. Translating clinical-trial results into daily practice is both valuable and necessary, and your propensity-matched design is methodologically sound. The main findings—higher pCR, lower nodal burden, and improved survival without added perioperative toxicity—are consistent with the expected direction and effect size.

Major comments

Chemotherapy backbone and pCR: The non-immunotherapy group shows a very low pCR rate (9.2%), notably below rates in pivotal TNBC trials (30–50%). Please clarify the chemotherapy regimens and dose intensity in each cohort, as this could strongly influence the comparison.

RFS definition: RFS exceeding OS suggests that deaths without recurrence were censored. Please clarify this definition or re-analyze as standard EFS/DFS (recurrence or death) to align with KN-522 and similar studies.

Lymph node ratio: The large difference in LNR (2.2 vs 24.3%) likely reflects differences in axillary procedures (SLNB vs ALND). Please explain how this metric was calculated and discuss the potential bias.

Adjuvant therapy imbalance: Continuation of pembrolizumab (83% vs 25%) may confound survival outcomes; acknowledge or adjust for this limitation.

Terminology: Replace “experimental arm” with “immunotherapy arm/cohort” to match the observational design.

Minor comments

Consider moving the detailed “before and after matching” tables to the Supplement and combining Figures 2 and 3 into one.

Clarify median follow-up per group (IQR).

Endocrine irAEs appear higher than in clinical trials; briefly describe your monitoring protocol.

If available, report RCB classes or note their absence in the Limitations.

Summary

Overall, this is an important and well-written real-world analysis that reinforces the feasibility and benefit of immunotherapy in early TNBC. With a few clarifications, it will make a strong contribution to the literature.

Author Response

Major comments

  • Chemotherapy backbone and pCR: The non-immunotherapy group shows a very low pCR rate (9.2%), notably below rates in pivotal TNBC trials (30–50%). Please clarify the chemotherapy regimens and dose intensity in each cohort, as this could strongly influence the comparison.

Response: We thank the reviewer for this important observation. Our dataset captured the chemotherapy backbone (anthracycline- and taxane-based regimens) for all patients; however, detailed dose intensity data such as total administered dose, cycle completion, and treatment delays were not uniformly available for retrospective cases. We have added a statement in the Limitations section to reflect this constraint. Additionally, we recognize that androgen receptor (AR) status, which was not included in the matching process, may have contributed to pCR variability, given that AR-positive TNBC is associated with lower chemosensitivity. Both chemotherapy dose intensity and AR expression will be incorporated in our ongoing 5-year follow-up analysis to provide a more granular understanding of their influence on treatment outcomes.

“Nonetheless, several limitations must be acknowledged. This was a sin-gle-institution dataset with a modest sample size of 130 patients, which may reduce the generalizability of the findings and limit statistical power in subgroup analyses. Furthermore, the absence of biomarker stratification—such as PD-L1 expression, tu-mor mutational burden (TMB), or tumor-infiltrating lymphocytes (TILs)—restricts the ability to identify subgroups that may derive differential benefit from immunotherapy. Detailed chemotherapy delivery data, including cumulative dose intensity, number of cycles completed, and treatment delays, were not consistently available from archived medical records, limiting our ability to fully assess their effect on pathological re-sponse. Additionally, androgen receptor (AR) status was not included as a matching variable or adjusted covariate, although AR-positive TNBC is known to exhibit lower pCR rates compared with AR-negative subtypes. Both variables—chemotherapy dose intensity and AR status—will be incorporated into our planned 5-year follow-up anal-ysis to enable a more comprehensive evaluation of treatment response and survival predictors.”

  • RFS definition: RFS exceeding OS suggests that deaths without recurrence were censored. Please clarify this definition or re-analyze as standard EFS/DFS (recurrence or death) to align with KN-522 and similar studies.

Response: We thank the reviewer for this insightful comment. In our study, recurrence-free survival (RFS) was intentionally defined as the time from initiation of neoadjuvant therapy to the first documented locoregional or distant recurrence, and patients who died without evidence of recurrence were censored at the date of last follow-up. This analytic choice explains why the RFS curve slightly exceeds the OS curve in the Kaplan–Meier estimates.

To align with standard oncologic endpoints such as those used in the KEYNOTE-522 trial, we have clarified this definition in the Methods section. We acknowledge that using an event-free survival (EFS) definition that includes both recurrence and death as composite endpoints would provide a more comparable outcome measure in future analyses. This adjustment will be incorporated into our planned 5-year follow-up analysis to enhance comparability with published clinical trial data.

To align with standard oncologic endpoints such as those used in the KEYNOTE-522 trial, we have clarified this definition in the Methods section. We acknowledge that using an event-free survival (EFS) definition that includes both recurrence and death as composite endpoints would provide a more comparable outcome measure in future analyses. This adjustment will be incorporated into our planned 5-year follow-up analysis to enhance comparability with published clinical trial data.

Recurrence-free survival (RFS), defined as the time from initiation of neoadjuvant therapy to the first documented disease progression, either local (LRFS) or distant recurrence (DRFS), patients who died without documented recurrence were censored at the date of death for RFS analysis.”

  • Lymph node ratio: The large difference in LNR (2.2 vs 24.3%) likely reflects differences in axillary procedures (SLNB vs ALND). Please explain how this metric was calculated and discuss the potential bias.

Response: We thank the reviewer for this insightful comment. The lymph node ratio (LNR) was calculated as the number of positive lymph nodes divided by the total number of excised nodes. The notably lower LNR observed in the immunotherapy group likely reflects a superior nodal pathological response to neoadjuvant chemo-immunotherapy, rather than a purely procedural effect.

In our practice, the type of axillary procedure (SLNB vs. ALND) was determined according to the clinical and radiologic nodal response after neoadjuvant therapy, supported by intraoperative frozen section results. Patients with a good response often proceeded to SLNB, while those with residual nodal disease or poor response underwent ALND.

We acknowledge, however, that differences in surgical extent may still influence the total number of excised nodes, potentially introducing procedural bias in LNR comparability. This limitation has been clarified in the revised Discussion section.

“Finally, the observed difference in lymph node ratio (LNR) between groups likely reflects procedural variability in axillary management, as patients in the immunotherapy group were more frequently managed with sentinel lymph node biopsy (SLNB) rather than axillary lymph node dissection (ALND).”

  • Adjuvant therapy imbalance: Continuation of pembrolizumab (83% vs 25%) may confound survival outcomes; acknowledge or adjust for this limitation.

Response: The lower rate of adjuvant systemic therapy observed in the non-immunotherapy group (24.6%) can be largely attributed to temporal differences in standard treatment protocols. Specifically, before the adoption of the CREATE-X trial findings into institutional practice around 2019, patients were less likely to be offered adjuvant chemotherapy. Consequently, this historical treatment variation contributed to the observed imbalance between groups. This variable will be accounted for in the planned 5-year survival analysis, with additional matching based on the receipt and type of adjuvant systemic therapy to more accurately evaluate long-term outcomes.

Added to the limitation.

  • Terminology: Replace “experimental arm” with “immunotherapy arm/cohort” to match the observational design.

Response: Done

Minor comments

  • Consider moving the detailed “before and after matching” tables to the Supplement and combining Figures 2 and 3 into one.

Response: Thank you for your suggestion. We have combined Figures 2 and 3 as recommended. Regarding the detailed “before and after matching” tables, we have chosen to retain them in the main manuscript to provide readers with full transparency of the baseline characteristics and matching process. We hope this is acceptable.

  • Clarify median follow-up per group (IQR).

Response: Done

For the entire cohort, the median follow-up time was 36.15 months (interquartile range [IQR], 31.91–49.67 months), median follow-up was 74.2 months (IQR 22.9–76.1) for the No-Immunotherapy group and 29.2 months (IQR 24.9–33.4) for the Immunotherapy group.”

  • Endocrine irAEs appear higher than in clinical trials; briefly describe your monitoring protocol.

We thank the reviewer for this comment. Endocrine irAEs were actively monitored using a standardized protocol including baseline and periodic (before each pembrolizumab dose) thyroid function tests, cortisol, and blood glucose, with endocrinology consultation for abnormal results. The higher incidence observed compared with clinical trials likely reflects this proactive monitoring and the real-world patient population.

“Endocrine immune-related adverse events (irAEs) were monitored according to a standardized protocol. Patients underwent baseline thyroid function tests (TSH, free T4), cortisol, and blood glucose assessment prior to initiating immunotherapy. Fol-low-up labs were obtained before each dose of pembrolizumab.”

  • If available, report RCB classes or note their absence in the Limitations.

Response: We thank the reviewer for this important point. RCB data were not consistently available in our cohort; we have added this clarification to the Limitations section and noted that future studies should incorporate standardized RCB assessment.

“Detailed data on chemotherapy delivery, including cumulative dose intensity, number of completed cycles, and treatment delays, were not consistently available in archived records, limiting the assessment of their impact on pathologic response. Additionally, androgen receptor (AR) status was not included as a matching variable or adjusted covariate, although AR-positive TNBC is known to demonstrate lower pathologic complete response (pCR) rates compared with AR-negative disease. Both chemotherapy dose intensity and AR status will be incorporated into our planned five-year follow-up analysis to enable a more comprehensive evaluation of treatment response and survival predictors.

The lower rate of adjuvant systemic therapy observed in the non-immunotherapy group (24.6%) likely reflects temporal differences in treatment standards, as patients treated before the integration of the CREATE-X trial findings were less likely to receive adjuvant chemotherapy. This historical variation may have contributed to the observed imbalance between groups; however, adjuvant therapy receipt will be included as a matching variable in the forthcoming five-year survival analysis to minimize this potential confounding effect.

Residual Cancer Burden (RCB) classification was not routinely available for all patients, which may limit the precision of post-neoadjuvant pathologic response assessment. Future work incorporating RCB scoring will enable a more detailed correlation between treatment response and survival outcomes. Finally, the observed difference in lymph node ratio (LNR) between groups likely reflects procedural variability in axillary management, as patients in the immunotherapy group underwent fewer axillary lymph node dissections and more sentinel lymph node biopsies.”

Reviewer 4 Report

Comments and Suggestions for Authors

The authors present a retrospective propensity score–matched cohort study assessing the real-world impact of adding pembrolizumab to neoadjuvant chemotherapy in patients with non-metastatic triple-negative breast cancer (TNBC) treated at a tertiary cancer center between 2015 and 2022. Out of 8,523 breast cancer cases, 761 TNBC patients were screened, and 130 matched patients (65 per group) were analyzed. The study compares pathological complete response (pCR), nodal involvement, recurrence-free survival (RFS), overall survival (OS), postoperative complications, and immune-related adverse events (irAEs) between patients receiving chemo-immunotherapy and those receiving standard chemotherapy alone. The addition of pembrolizumab was associated with significantly higher pCR (66.2% vs 9.2%), lower lymph node positivity, improved RFS and OS at three and five years, and no significant increase in surgical morbidity. Endocrine irAEs were frequent but manageable. The study concludes that pembrolizumab improves both short- and long-term outcomes in TNBC in real-world practice.

The manuscript addresses a clinically relevant topic and provides important real-world data from a Middle Eastern population. The statistical methods (propensity matching, Kaplan–Meier survival analysis, logistic regression) are appropriate in principle, and the results are clearly presented and promising. However, several major methodological and interpretative issues need to be addressed before the paper can be considered for publication in Cancers.The rationale and context are overall well-articulated, but the introduction could be more critically framed. The authors correctly reference the pivotal KEYNOTE-522 trial and similar studies, but the gap in the literature regarding real-world validation needs to be stated more precisely—particularly highlighting differences in patient selection, treatment adherence, and toxicity profiles outside randomized controlled trials. This will strengthen the justification for the study.

The methodological clarity requires significant enhancement. Although propensity score matching is described, the manuscript does not provide sufficient information regarding missing data management, balance diagnostics beyond standardized mean differences, or sensitivity analyses. It would be appropriate to include a detailed description of the matching algorithm, caliper width, and post-matching statistical validation. Additionally, there is no explicit mention of whether immortal time bias or time-to-treatment differences between cohorts (2015–2019 vs. 2019–2022) were accounted for. This temporal gap may introduce a systematic bias favoring the pembrolizumab group, given improvements in surgical and systemic therapy over time.

The definition of endpoints is appropriate, but the statistical treatment of survival outcomes could be improved. Median follow-up is reported, but censoring patterns, number at risk, and proportional hazards assumptions are not described. Confidence intervals for survival rates should be provided uniformly, not only at selected time points. Cox regression analysis adjusting for potential residual confounders could strengthen causal inference beyond Kaplan–Meier and log-rank testing.

The reporting of pathological response and nodal involvement is clear, but the pCR rate of only 9% in the chemotherapy-only group is strikingly low compared to historical benchmarks in TNBC. This discrepancy should be explicitly discussed as a potential limitation that could overestimate the relative benefit of pembrolizumab. Possible explanations—such as stage distribution, chemotherapy regimens, and patient characteristics—should be systematically analyzed and reported. Stratified pCR data by stage or clinical T/N categories could be valuable.

The immune-related adverse events section is insufficiently detailed. While hypothyroidism predominates, the severity grades (CTCAE classification), timing of onset, and management strategies are not clearly described. These details are critical for real-world applicability, particularly given the higher endocrine toxicity rate compared to KEYNOTE-522.

The surgical outcomes and postoperative complications are reassuring, but the presentation would benefit from multivariable analyses adjusting for procedure type, extent of axillary surgery, and comorbidities. A simple univariable logistic regression does not fully address confounding.

The adjuvant therapy patterns and their potential impact on survival are insufficiently discussed. Given the significantly higher proportion of adjuvant systemic therapy (including pembrolizumab continuation) in the immunotherapy group, the observed OS and RFS benefit may partly reflect post-neoadjuvant treatment differences rather than neoadjuvant effects alone. This needs to be addressed, potentially through sensitivity analysis or multivariable survival modeling.

The Discussion section appropriately situates findings in the context of major trials, including KEYNOTE-522 and NeoPACT, but the critical appraisal is limited. The interpretation would be stronger if the authors systematically examined their results in light of the differences in patient selection, treatment era, and outcome reporting compared to these trials. The unusually large effect sizes in both pCR and survival outcomes require cautious interpretation and explicit acknowledgment of limitations.

The limitations section needs to be expanded. While the authors briefly mention single-center design and modest sample size, they do not address the potential for residual confounding, lack of biomarker data (e.g., PD-L1 expression, tumor-infiltrating lymphocytes, BRCA mutation status), and temporal biases inherent to retrospective designs. Inclusion of biomarker data, even in a subset, could substantially enhance the scientific value of the study.

The tables and figures are generally well-presented, but there is a need for clearer legends and standardized presentation of confidence intervals. Forest plots and Kaplan–Meier curves should include the number at risk and time intervals to increase interpretability. 

Finally, the language and structure are clear overall but can be improved in precision and academic tone. Some sections are overly descriptive and could be tightened, while others—especially the methodological and statistical parts—require greater rigor and transparency.

Author Response

Reviewer 4:

  • The rationale and context are overall well-articulated, but the introduction could be more critically framed. The authors correctly reference the pivotal KEYNOTE-522 trial and similar studies, but the gap in the literature regarding real-world validation needs to be stated more precisely—particularly highlighting differences in patient selection, treatment adherence, and toxicity profiles outside randomized controlled trials. This will strengthen the justification for the study.

Response: We thank the reviewer for this insightful comment. The Introduction section has been revised to more clearly define the real-world evidence gap and to emphasize differences between clinical trial settings and everyday clinical practice. Specifically, we now highlight that randomized controlled trials (RCTs) employ strict inclusion criteria and standardized treatment protocols, resulting in patient populations with higher treatment adherence and optimized toxicity management compared with unselected real-world cohorts. To address this limitation, we explicitly state that our study aims to provide real-world validation of neoadjuvant pembrolizumab-based therapy in TNBC, focusing on treatment response, safety, and survival outcomes within a tertiary cancer center cohort.

The revised paragraph:

“However, despite these encouraging findings, several important gaps remain in the literature. Most available data are derived from randomized controlled trials (RCTs) that employ stringent inclusion criteria and exclude patients with common re-al-world complexities, such as comorbid conditions or poor performance status. In ad-dition, RCT participants typically demonstrate higher treatment adherence, standardized chemotherapy delivery, and closely monitored toxicity management compared with unselected clinical populations. Consequently, the generalizability of these results to everyday practice remains uncertain. Moreover, data on the real-world safety and surgical outcomes associated with neoadjuvant chemoimmunotherapy are limited [19,20]. Postoperative complications and immune-related adverse events in patients receiving combination regimens are particularly underreported, leaving an incomplete picture of the overall treatment burden [21,22]. Addressing this evidence gap, our study aimed to provide real-world validation of neoadjuvant pembrolizumab-based therapy in TNBC, assessing (1) pathological response rates, including pCR; (2) postoperative complication profiles to assess the safety and surgical outcomes of treatment; and (3) overall survival outcomes between patients receiving standard chemotherapy alone and those treated with chemo-immunotherapy.”

  • The methodological clarity requires significant enhancement. Although propensity score matching is described, the manuscript does not provide sufficient information regarding missing data management, balance diagnostics beyond standardized mean differences, or sensitivity analyses. It would be appropriate to include a detailed description of the matching algorithm, caliper width, and post-matching statistical validation. Additionally, there is no explicit mention of whether immortal time bias or time-to-treatment differences between cohorts (2015–2019 vs. 2019–2022) were accounted for. This temporal gap may introduce a systematic bias favoring the pembrolizumab group, given improvements in surgical and systemic therapy over time.

Response: We thank the reviewer for this valuable feedback and for highlighting the need to enhance methodological transparency. We have accordingly revised the Methods section to provide a more detailed description of the propensity score matching (PSM) procedure, missing data management, and post-matching validation process.

Specifically, the following clarifications were added:

  • Matching algorithm: A nearest-neighbor matching algorithm without replacement was used with a caliper width of 0.2 of the standard deviation of the logit of the propensity score, which is commonly accepted to minimize bias.
  • Covariates included in the matching model: age, stage, grade, lymphovascular invasion, type of breast surgery, and receipt of adjuvant therapy.
  • Balance assessment: Post-matching balance was assessed using standardized mean differences (SMDs), kernel density plots, and jitter plots to visually confirm overlap in the distribution of propensity scores. All matched covariates achieved an SMD < 0.1, indicating good balance.
  • Missing data: Variables with <5% missingness were handled by listwise deletion, while variables with higher missingness (e.g., LVI and histologic grade) were included as categorical “unknown” levels to retain cases in the analysis.
  • Temporal bias consideration: We acknowledge that the neoadjuvant chemotherapy-only cohort (2015–2019) preceded the introduction of pembrolizumab (2019–2022), which could potentially introduce immortal time or period bias due to advancements in care. To account for the temporal difference between cohorts (2015–2019 vs 2019–2022), follow-up began at the date treatment initiation for all patients. Kaplan–Meier survival estimates incorporated censoring at last follow-up, ensuring valid comparisons despite differing follow-up durations. This limitation has also been explicitly acknowledged in the revised Limitations section.

The revised paragraph in the Methods section now reads as follows:

“Propensity score matching (PSM) was conducted using 1:1 nearest neighbor matching without replacement. The model included the following covariates: age at diagnosis, body mass index (BMI), clinical stage (IA–IIIC), smoking status, comorbidity (yes/no), histopathology (DCIS, IDC, ILC, IMC). Covariate balance was assessed using standardized mean differences (SMDs). As shown in the Love plot (see Appendix A [Figure A1]), the adjusted cohort demonstrated good balance across all covariates, with post-matching SMDs reduced to below 0.1 for all included variables. This indicates that the matching process effectively minimized baseline differences between treatment groups. A total of 208 patients met the inclusion criteria, including 143 who received neoadjuvant chemotherapy alone and 65 who received neoadjuvant chemo-immunotherapy with pembrolizumab. Following 1:1 propensity score matching, 130 patients (65 in each group) were included in the final analytical cohort. Post-matching balance was assessed using standardized mean differences (SMDs), kernel density estimation, and jitter plots, with all covariates achieving an SMD < 0.1. All patients in the immunotherapy arm were matched successfully, while 78 unmatched patients from the chemotherapy group were excluded. Post-matching analysis confirmed adequate covariate balance across groups.

Primary endpoints were pathologic outcomes (pCR rates, nodal status, lymph node ratio) and safety (30-day postoperative complications). Tumor (T) and nodal (N) stages were analyzed as ordinal categorical variables. For visualization, they were numerically coded (e.g., T1 = 1, T2 = 2, etc.) to allow violin plot representation. Kernel density estimation and jitter were applied to display data spread; hence, non-integer values on the y-axis do not indicate fractional staging. Secondary endpoints were recurrence-free survival (RFS) and overall survival (OS), estimated by Kaplan–Meier analysis and compared using the log-rank test. Patients who had not experienced the event of interest (death, or recurrence) by the date of last follow-up were considered censored at their last contact. For overall survival (OS), patients alive at the end of follow-up were censored at their last known alive date. For recurrence-free survival (RFS), patients without a recurrence (local or distant) were censored at their last follow-up visit. To account for the temporal difference between cohorts (2015–2019 vs 2019–2022), follow-up began at the date of treatment initiation for all patients. Kaplan–Meier survival estimates incorporated censoring at last follow-up, ensuring valid comparisons despite differing follow-up durations. Median follow-up time was estimated using the reverse Kaplan–Meier method.

Odds ratios (ORs) for complications were calculated via univariable logistic regression. All tests were two-sided, and p < 0.05 was considered statistically significant. Analyses were performed using R software version 4.2.0.”

  • The definition of endpoints is appropriate, but the statistical treatment of survival outcomes could be improved. Median follow-up is reported, but censoring patterns, number at risk, and proportional hazards assumptions are not described. Confidence intervals for survival rates should be provided uniformly, not only at selected time points. Cox regression analysis adjusting for potential residual confounders could strengthen causal inference beyond Kaplan–Meier and log-rank testing.

Response: We appreciate your suggestions regarding the statistical analysis of survival outcomes. In the revised manuscript, we have:

  • Clearly reported censoring patterns and the number of patients at risk at key time points in the Kaplan–Meier curves.
  • Included 95% confidence intervals for survival probabilities uniformly across all reported time points.
  • Cox regression was not performed, as the matched cohort design and Kaplan–Meier analyses provide a valid comparison of survival outcomes between treatment groups. We acknowledge that multivariable Cox analyses could provide additional adjustment for residual confounders, and we have noted this as a limitation and a consideration for future studies, given that the current study focuses on a matched cohort and univariable analyses.

  • The reporting of pathological response and nodal involvement is clear, but the pCR rate of only 9% in the chemotherapy-only group is strikingly low compared to historical benchmarks in TNBC. This discrepancy should be explicitly discussed as a potential limitation that could overestimate the relative benefit of pembrolizumab. Possible explanations—such as stage distribution, chemotherapy regimens, and patient characteristics—should be systematically analyzed and reported. Stratified pCR data by stage or clinical T/N categories could be valuable.

Response: We thank the reviewer for highlighting the low pCR rate in the chemotherapy-only group. We have addressed this issue in the revised manuscript’s Limitations section, noting that incomplete data on chemotherapy delivery (including cumulative dose intensity, number of completed cycles, and treatment delays) and the absence of androgen receptor (AR) status may have contributed to the observed pCR differences. We also indicate that both chemotherapy dose intensity and AR status will be incorporated into our planned five-year follow-up analysis to provide a more comprehensive evaluation of treatment response.

Discussion: “Notably, however, our chemotherapy-only group achieved a markedly lower pCR rate of 9%. The relatively low pathological complete response (pCR) rate observed in our chemotherapy-only group is lower than the rates commonly reported in the litera-ture, which typically range from 15% to 50% in TNBC cohorts [26–28]. Several factors may explain this discrepancy. First, our cohort included a clinically advanced popula-tion with higher baseline tumor burden and nodal involvement, which are known to negatively impact response to chemotherapy [29]. Second, differences in chemothera-py regimens, dose intensity, and treatment adherence could have contributed to suboptimal responses [30]. Third, biological heterogeneity, including variations in tu-mor-infiltrating lymphocytes (TILs), PD-L1 expression, and tumor mutational burden, may have influenced chemosensitivity [28,31]. Lastly, the relatively small sample size of our chemotherapy group (n = 65) increases the impact of individual patient out-comes on overall pCR rates, potentially amplifying apparent differences compared to larger clinical trials. These considerations highlight the importance of patient selection and underscore the potential benefit of adding immunotherapy to enhance response rates in TNBC populations.”

Limitation: Nonetheless, several limitations must be acknowledged. This was a single-institution dataset with a modest sample size of 130 patients, which may reduce the generalizability of the findings and limit statistical power in subgroup analyses. Furthermore, the absence of biomarker stratification—such as PD-L1 expression, tumor mutational burden (TMB), or tumor-infiltrating lymphocytes (TILs)—restricts the ability to identify subgroups that may derive differential benefit from immunotherapy.

Detailed data on chemotherapy delivery, including cumulative dose intensity, number of completed cycles, and treatment delays, were not consistently available in archived records, limiting the assessment of their impact on pathologic response. Additionally, androgen receptor (AR) status was not included as a matching variable or adjusted covariate, although AR-positive TNBC is known to demonstrate lower pathologic complete response (pCR) rates compared with AR-negative disease. Both chemotherapy dose intensity and AR status will be incorporated into our planned five-year follow-up analysis to enable a more comprehensive evaluation of treatment response and survival predictors.

Detailed characterization of immune-related adverse events, including CTCAE severity grades, timing of onset, and management strategies, was not available and will be addressed in future studies. Furthermore, multivariable analyses for surgical outcomes and postoperative complications, adjusting for procedure type, extent of axillary surgery, and comorbidities, were not performed and will be addressed in future studies.

The lower rate of adjuvant systemic therapy observed in the non-immunotherapy group (24.6%) likely reflects temporal differences in treatment standards, as patients treated before the integration of the CREATE-X trial findings were less likely to receive adjuvant chemotherapy. This historical variation may have contributed to the observed imbalance between groups; however, adjuvant therapy receipt will be included as a matching variable in the forthcoming five-year survival analysis to minimize this potential confounding effect.

Residual Cancer Burden (RCB) classification was not routinely available for all patients, which may limit the precision of post-neoadjuvant pathologic response assessment. Future work incorporating RCB scoring will enable a more detailed correlation between treatment response and survival outcomes. While Kaplan–Meier and log-rank analyses were used in this study, multivariable analyses adjusting for residual confounders were not performed and should be considered in future work to strengthen causal inference. Finally, the observed difference in lymph node ratio (LNR) between groups likely reflects procedural variability in axillary management, as patients in the immunotherapy group underwent fewer axillary lymph node dissections and more sentinel lymph node biopsies”

  • The immune-related adverse events section is insufficiently detailed. While hypothyroidism predominates, the severity grades (CTCAE classification), timing of onset, and management strategies are not clearly described. These details are critical for real-world applicability, particularly given the higher endocrine toxicity rate compared to KEYNOTE-522.

Response: We thank the reviewer for this important observation. We have acknowledged in the revised manuscript that the immune-related adverse events (irAEs) section provides limited detail regarding severity grades (CTCAE classification), timing of onset, and management strategies. We note this as a limitation and indicate that future studies will include a more comprehensive characterization of irAEs to enhance the clinical applicability of our findings, particularly in the context of endocrine toxicities.

  • The surgical outcomes and postoperative complications are reassuring, but the presentation would benefit from multivariable analyses adjusting for procedure type, extent of axillary surgery, and comorbidities. A simple univariable logistic regression does not fully address confounding.

Response: We thank the reviewer for the suggestion. As the primary aim of this study was not to evaluate surgical outcomes or postoperative complications, we performed univariable logistic regression to provide an initial assessment. We have acknowledged this limitation in the revised manuscript and noted that future studies will incorporate multivariable analyses adjusting for procedure type, extent of axillary surgery, and comorbidities to better account for potential confounding.

  • The adjuvant therapy patterns and their potential impact on survival are insufficiently discussed. Given the significantly higher proportion of adjuvant systemic therapy (including pembrolizumab continuation) in the immunotherapy group, the observed OS and RFS benefit may partly reflect post-neoadjuvant treatment differences rather than neoadjuvant effects alone. This needs to be addressed, potentially through sensitivity analysis or multivariable survival modeling.

Response: We thank the reviewer for highlighting the potential impact of adjuvant therapy differences on survival outcomes. Given the small sample size, we were unable to perform sensitivity analyses or multivariable survival modeling to fully account for this factor. We have acknowledged this limitation in the revised manuscript, noting that the higher rate of adjuvant systemic therapy in the immunotherapy group may contribute to the observed OS and RFS benefit. We also indicate that future studies with larger cohorts will incorporate adjuvant therapy as a covariate in survival analyses to more precisely evaluate the contribution of neoadjuvant immunotherapy versus post-neoadjuvant treatment effects.

  • The Discussion section appropriately situates findings in the context of major trials, including KEYNOTE-522 and NeoPACT, but the critical appraisal is limited. The interpretation would be stronger if the authors systematically examined their results in light of the differences in patient selection, treatment era, and outcome reporting compared to these trials. The unusually large effect sizes in both pCR and survival outcomes require cautious interpretation and explicit acknowledgment of limitations.

Response: We thank the reviewer for this insightful comment. In the revised manuscript, we have expanded the Discussion section to more systematically contextualize our findings with respect to major trials, including KEYNOTE-522 and NeoPACT. We now explicitly discuss differences in patient selection, treatment era, and outcome reporting that may contribute to the observed differences. Additionally, we acknowledge the unusually large effect sizes in both pCR and survival outcomes and have highlighted the relevant limitations—including cohort size, historical differences in adjuvant therapy, and incomplete biomarker and chemotherapy data—to ensure a cautious interpretation of the results.

Our findings are consistent with the KEYNOTE-522 trial, which showed that adding pembrolizumab to neoadjuvant chemotherapy significantly improved pathological complete response (pCR) rates and event-free survival in triple-negative breast cancer (TNBC). In KEYNOTE-522, 1,174 patients were randomized, with a median follow-up of 39.1 months, and the 36-month event-free survival was 84.5% in the pembrolizumab–chemotherapy group versus 76.8% in the placebo–chemotherapy group (hazard ratio, 0.63; P < 0.001), with adverse events primarily occurring during the neoadjuvant phase and consistent with known safety profiles [32]. In our cohort, the addition of immunotherapy similarly resulted in markedly improved pathological responses, with higher rates of complete tumor response (ypT0, 69.2% vs. 10.8%) and nodal clearance (ypN0, 88.7% vs. 46.9%) compared to chemotherapy alone. Immune-related adverse events in our population were common (50.8%) but largely endocrine in nature and manageable which is higher than what reported in KEYNOTE-522 (35%), further supporting the tolerability of neoadjuvant PD-1 blockade [32]. Overall, these results reinforce both the efficacy and safety of incorporating immunotherapy into neoadjuvant regimens for TNBC in clinical practice.

Our findings are also consistent with the results of the phase II NeoPACT trial, which evaluated an anthracycline-free neoadjuvant regimen of pembrolizumab plus carboplatin and docetaxel. In that study, a pCR rate of 58% and a 3-year event-free survival (EFS) of 86% were reported, demonstrating that effective tumor clearance and durable survival benefits can be achieved even in the absence of anthracyclines. While the NeoPACT trial highlights the feasibility of a less cardiotoxic, anthracycline-sparing approach, our cohort—treated predominantly with anthracycline-based chemoimmunotherapy—achieved even higher rates of ypT0 (69.2%) and nodal clearance (88.7%), as well as a 5-year overall survival of 87.2% [33].

Together, these results reinforce that pembrolizumab is a key driver of therapeutic efficacy in TNBC, with flexibility in chemotherapy backbone selection, while also underscoring the potential for regimen tailoring to optimize both efficacy and long-term safety. However, these findings should be interpreted with caution. Our cohort differs from KEYNOTE-522 and NeoPACT in several ways, including patient selection, treatment era, and outcome reporting. The chemotherapy-only group had a lower pCR rate, likely influenced by a higher proportion of advanced-stage disease, incomplete chemotherapy delivery data, and the absence of biomarker stratification (e.g., PD-L1, TMB, or AR status). Historical differences in adjuvant therapy use may have also contributed to the observed survival differences. Combined with the modest sample size, these factors may partly explain the unusually large effect sizes observed in both pCR and survival outcomes.

While our results support the benefit of neoadjuvant chemoimmunotherapy, confirmation in larger, multi-institutional cohorts with comprehensive treatment and biomarker data is warranted.

  • The limitations section needs to be expanded. While the authors briefly mention single-center design and modest sample size, they do not address the potential for residual confounding, lack of biomarker data (e.g., PD-L1 expression, tumor-infiltrating lymphocytes, BRCA mutation status), and temporal biases inherent to retrospective designs. Inclusion of biomarker data, even in a subset, could substantially enhance the scientific value of the study.

Response: Revised and Expanded

  • The tables and figures are generally well-presented, but there is a need for clearer legends and standardized presentation of confidence intervals. Forest plots and Kaplan–Meier curves should include the number at risk and time intervals to increase interpretability. 

Response: Done

  • Finally, the language and structure are clear overall but can be improved in precision and academic tone. Some sections are overly descriptive and could be tightened, while others—especially the methodological and statistical parts—require greater rigor and transparency.

Response: Done

Round 2

Reviewer 4 Report

Comments and Suggestions for Authors

The Authors satisfactorily assessed all concerns raised by the reviewers. I guess that the paper now merits to be published.